# Deposition of Nanocrystalline Multilayer Graphene Using Pulsed Laser Deposition

Yuxuan Wang, Bin Zou, Bruno Rente, Neil Alford and Peter K. Petrov *

Department of Materials, Imperial College London, London SW7 2AZ, UK
* Correspondence: p.petrov@imperial.ac.uk

**Abstract:** The wide application of graphene in the industry requires the direct growth of graphene films on silicon substrates. In this study, we found a possible technique to meet the requirement above. Multilayer graphene thin films (MLG) were grown without a catalyst on Si/SiO$_2$ using pulsed laser deposition (PLD). It was found that the minimum number of laser pulses required to produce fully covered (uninterrupted) samples is 500. This number of laser pulses resulted in samples that contain ~5 layers of graphene. The number of layers was not affected by the laser fluence and the sample cooling rate after the deposition. However, the increase in the laser fluence from 0.9 J/cm$^2$ to 1.5 J/cm$^2$ resulted in a 2.5-fold reduction in the MLG resistance. The present study reveals that the PLD method is suitable to produce nanocrystalline multilayer graphene with electrical conductivity of the same magnitude as commercial CVD graphene samples.

**Keywords:** graphene; pulsed laser deposition; nanocrystalline materials





## 1. Introduction

Since its discovery in 2004, graphene has gained more and more attention from various research areas. However, the wide application of single-layer graphene is still challenging due to the low efficiency of traditional mechanical exfoliations and the low grain size of graphene produced by CVD methods. Multilayer graphene attains a better trade-off between expense and physical property. Furthermore, the unconventional superconductivity of bilayer graphene with magic angle has been discovered recently [1]. Thus, developing a quick and simple method to grow multilayer graphene on the desired substrate is essential.

Graphene is a new allotrope of carbon. Having been discovered by KS Novoselov and AK Geim [2] in 2004, graphene has attracted plenty of attention from various fields. Graphene is noted for its high carrier mobility [3], high light transparency of 97%, the realization of room temperature quantum Hall effect [4], and high Young's modulus of 1 TPa [5].

Although many different growth methods have been developed since the discovery of graphene, CVD and mechanical exfoliation are still the two most commonly used methods. However, CVD methods can only facilitate the growth of graphene on metal foils, which requires a further transfer process and can introduce contamination and additional costs. By using mechanical exfoliation, it is possible to produce high-quality graphene thin film, but the method has a sample size limitation and is very time-consuming.

Moreover, single-layer graphene is not required for many practical applications such as electrical conducting materials. A method that can directly grow multilayer graphene layers on the silicon substrates is required. As a widely used physical vapor deposition method, PLD has been used to grow ceramic thin films for several decades.

In the last two decades, many attempts to obtain graphene thin films by using the PLD method have already been made. Many groups across the world obtained multilayer graphene via direct growth on various substrates such as Si/SiO$_2$ and sapphire [6]. However, few people explored the type of produced graphene and the effect of the parameter on

the quality of the graphene thoroughly. It was already found that the laser fluence and the number of pulses can affect the quality and the number of layers of graphene. However, the effect of the cooling rate on the electrical property of graphene grown by the PLD method is rarely examined.

The main topic of the reported research is the impact of the PLD parameters on graphene electrical properties, while the crystallinity is mainly under investigated. Jeong Hun Kwak et al. [7] reported the nanocrystalline structure of the graphene grown using the PLD method. However, the relation between the FWHM of the Raman spectrum peaks of nanocrystalline graphene and the resistance is still absent.

This paper explores the deposition process of multilayer graphene using the PLD method and presents the influence of laser fluence and the cooling rate after the deposition on the properties of the samples.

## 2. Materials and Methods

The process of graphene growth by using the PLD method was performed as follows: $1 \times 1$ cm$^2$ Si/SiO$_2$ substrates with 90 nm oxide layers were mounted on the substrate holder first, and then the substrates were heated to 800 °C at the rate of 50 °C/min in the vacuum level of $3 \times 10^{-5}$ Torr. A total of 500 pulses of KrF laser (248 nm) with the frequency of 10 Hz ablated the graphite target (C 009600 Goodfellow) for the graphene deposition. The used laser fluences were 0.9, 1.1, and 1.5 J/cm$^2$. Once the deposition was finished, the samples were cooled at 5, 10, 20, 30, 50, 70, and 90 °C/min to 305 °C, after which natural cooling was applied to room temperature.

All XPS spectra were recorded using a K-alpha+ XPS spectrometer equipped with a MXR3 Al K$\alpha$ monochromated X-ray source (h$\nu$ = 1486.6 eV). X-ray gun power was set to 72 W (6 mA and 12 kV). With this X-ray setting, the intensity of the Ag 3d5/2 photoemission peak for an atomically clean Ag sample, recorded at a pass energy (PE) of 20 eV, was $5 \times 10^6$ counts s$^{-1}$, and the full width at half maximum (FWHM) was 0.58 eV. Binding energy calibration was made using Au 4f7/2 (83.96 eV), Ag 3d5/2 (368.21 eV), and Cu 2p3/2 (932.62 eV). Charge compensation was achieved using the FG03 flood gun using a combination of low-energy electrons and the ion flood source.

Raman spectroscopy measurements were performed using a LabRAM HR Evolution HORIBA Raman spectrometer with a laser wavelength of 532 nm (excitation energy EL = $\hbar$wL = 2.33 eV), which used an optical fiber, an objective lens of 100$\times$, and NA = 0.8, resulting in a laser spot of 0.4 μm. The laser power was kept below 2 mW, and the spectral resolution was $\sim$3 cm$^{-1}$; the Raman peak position was calibrated based on the Si peak position at 520.7 cm$^{-1}$.

The Asylum Research MFP-3D AFM system was used to characterize the topology of graphene samples and the thickness of the multilayer graphene. The type of AFM measurement chosen was AC air topography.

A total of 16 Au (50 nm)/Ti (5 nm) electrodes with a diameter of 1 mm were deposited on the surface of the graphene samples via magnetron sputtering. The resistance across the graphene samples was measured using a probe station (Model: Signatone S-1160) and a Keysight B1500 Semiconductor Analyzer.

## 3. Results and Discussion

X-ray photoelectron spectroscopy was used to further characterize the samples. The result is shown in Figure 1 from which C1s, O1s, and Si2p are characterized. All the curves are plotted and fitted in Origin with the Voigt function. The C1s peak can be seen as the sum of the sp$^2$ carbon, C-O, C=O, and C-H. In both samples, the sp$^2$ carbon takes the highest portion, which means that both samples are graphene. However, the proportion of the signal from the other types of carbon atoms in graphene grown with a laser fluence of 0.9 J/cm$^2$ is higher than that in the graphene grown with 1.5 J/cm$^2$. This means that the quality of the graphene improves with the increase in the laser fluence. Additionally, the

portion of C-O in the O1s signal from the graphene sample grown with 0.9 J/cm$^2$ fluence is also higher than the one grown with 1.5 J/cm$^2$. This further confirms the drawn conclusion.

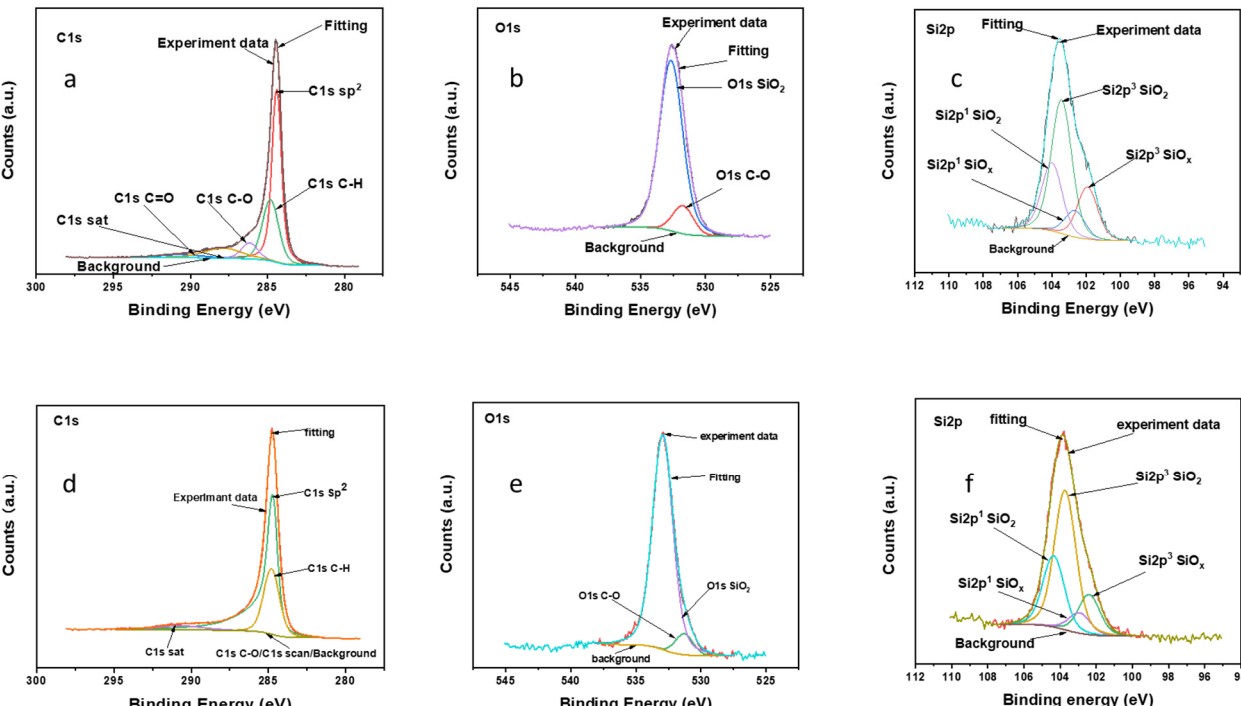

**Figure 1.** The XPS result of the graphene sample grown at the cooling rate of 50 °C/min under 0.9 J/cm$^2$ (**a–c**) and 1.5 J/cm$^2$ (**d–f**). The C1s, O1s, and Si2p of these two samples are shown above. The C1s result of both samples shows that most carbon atoms in the samples are sp$^2$ hybridized, which indicates that the sample obtained is graphene.

The Raman spectra of the multilayer graphene grown on the Si/SiO$_2$ substrate using the PLD method with laser fluences of 0.9 J/cm$^2$ and 1.5 J/cm$^2$ are shown in Figure 2. In both Raman spectra, the D peak is higher than the G peak, and the 2D peak is highly broadened with lower intensity compared to the G peak. According to Wu Jiang-Bin et al. [8], this indicates that the obtained graphene is polycrystalline with a nano-sized grain. This also explains the broad 2D peak and high D peak, since a large number of grain boundaries increases the graphene defect density.

To explore the effect of the cooling rate on the quality of the graphene, the multilayer graphene samples, after the deposition, were cooled down at a rate of 5, 10, 20, 30, 70, and 90 °C/min. The laser fluence was kept at 0.9 J/cm$^2$. It was observed that the Raman spectra are very similar and not affected by the cooling rates. This means that the nanocrystalline structure of the graphene samples is not influenced by the cooling rate. All the whole Raman spectra of all the samples are plotted and shown in Figure 3.

Another parameter that can affect the quality of the graphene is the number of layers. Traditionally, only carbon thin films with the number of layers in the range of 5–10 can be seen as multilayer graphene, while graphene with a number of layers less than five can be called few-layer graphene. The number of layers could be evaluated using Equation (1), derived by Bayle, Maxime et al. [9].

$$N_G = 1.05 \times A_G^{norm} + 0.16 \times (A_G^{norm})^2 \tag{1}$$

where $A_G^{norm}$ is the ratio between the G peak area of the graphene sample and the G peak area of the graphite target.

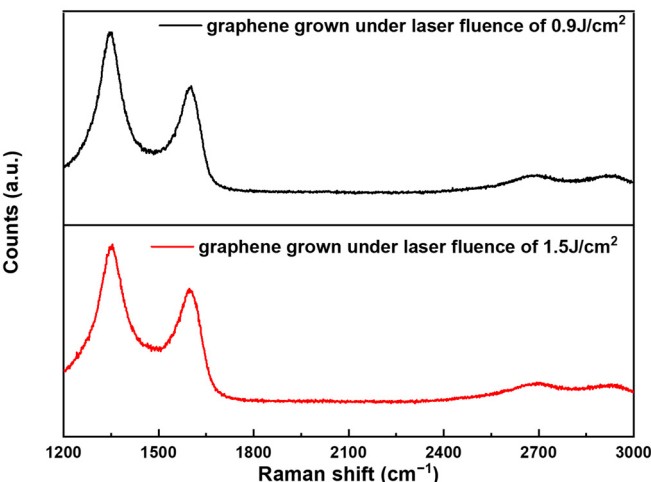

**Figure 2.** The Raman spectrum of multilayer graphene grown on Si/SiO$_2$ substrates using the PLD method at the cooling rate of 50 °C/min under 0.9 and 1.5 J/cm$^2$, respectively. The Raman spectrum confirms that the thin film grown using the PLD method is nanocrystalline graphene with multilayers.

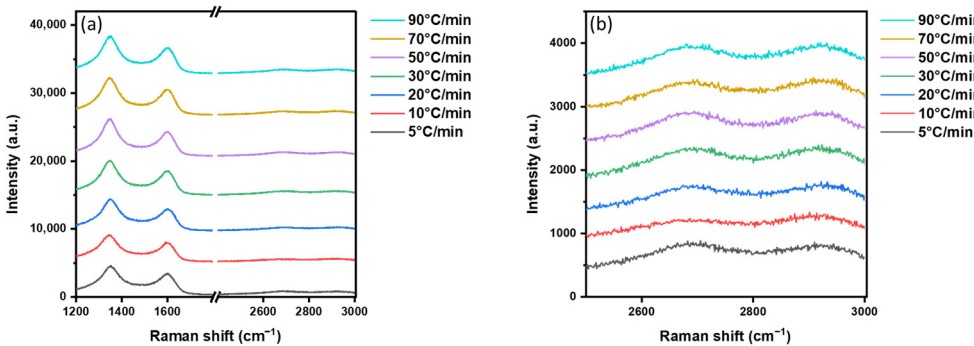

**Figure 3.** (**a**) The Raman spectra of nanocrystalline multilayer graphene grown using the PLD method with different cooling rates. (**b**) The broadened 2D peak inside the Raman spectra of graphene samples grown at different cooling rates.

The number of graphene layers evaluated for all samples is presented in Figure 4. It is around five, and not affected by either the laser beam fluence or the sample cooling rate. This shows that the difference in graphene quality is not aroused by the number of layers.

The verification of the above estimation was performed by characterization using AFM on the edges of the graphene samples grown with a laser fluence of 0.9 J/cm$^2$ at the cooling rates of 50 °C/min. The typical AFM image is shown in Figure 5. The average thickness of this graphene sample is around 1.5–2 nm. Since the thickness of single-layer graphene is 0.335 nm, the number of layers of the graphene sample is about 5–6.

To evaluate the quality of the graphene sample, the electrical resistance of the graphene samples was measured using four probe measurement methods. The result is shown in Figure 6. Evidently, the resistance of the graphene samples is affected by the laser fluence. By fixing the cooling rate at 50 °C/min, the resistance of the graphene was ~15 kΩ for the samples ablated with a 0.9 J/cm$^2$ laser energy, and ~6.6 kΩ when the laser energy was 1.5 J/cm$^2$. This result confirms the conclusion based on the XPS result that the graphene samples grown with a laser fluence of 1.5 J/cm$^2$ have better quality.

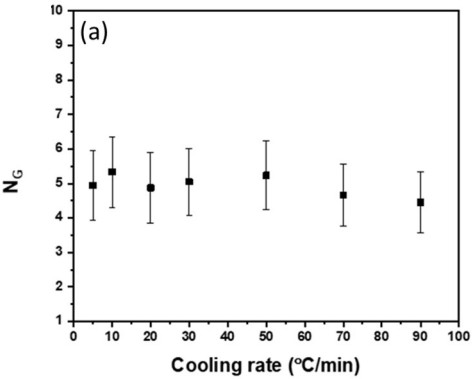 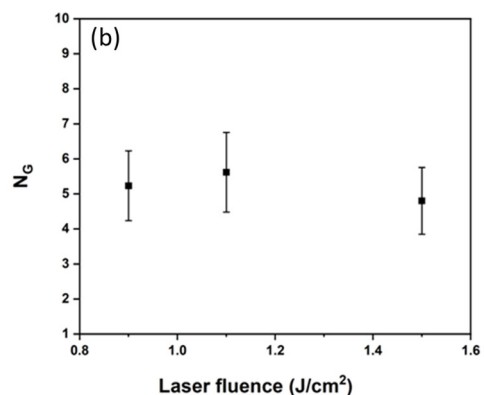

**Figure 4.** (**a**) The calculated number of layers of graphene grown under different cooling rates and (**b**) different laser fluences The points in the figure represent the calculated value of the number of layers of graphene samples at a different cooling rate based on Equation (1) and the error bars are the standard deviation of the number of layers due to area estimation. The points in (**b**) represent the calculated value of the number of layers of graphene samples at different laser fluences based on Equation (1) and the error bars are the standard deviation of the number of layers due to area estimation.

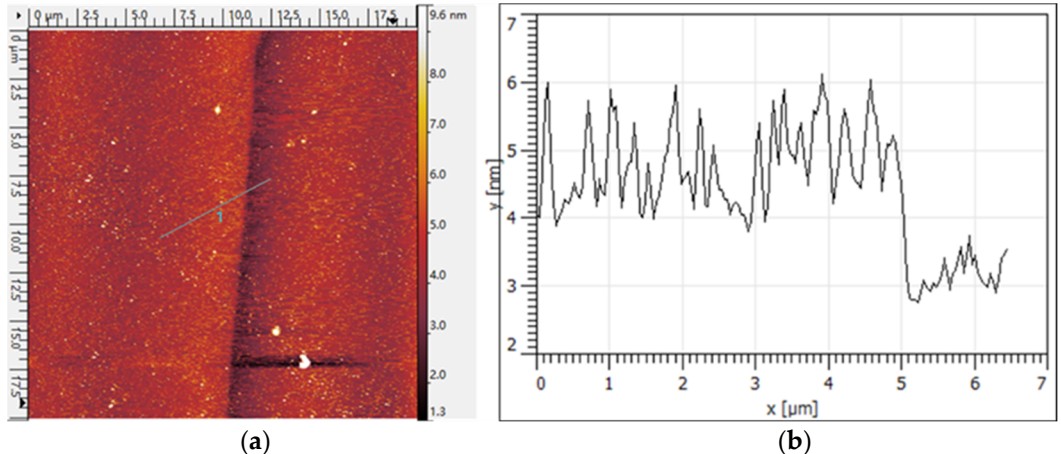

(**a**)                                        (**b**)

**Figure 5.** (**a**) The AFM image of the graphene sample grown under $0.9 \, \text{J/cm}^2$ at the cooling rate of $50 \, ^\circ\text{C/min}$. (**b**) The profile was plotted across the image. The profile shows that the thickness of the graphene sample is about 1.5–2 nm, which corresponds to 5–6 layers. The green line with '1' is the profile measured by AFM across the edge of the graphene sample on the $Si/SiO_2$ substrate.

  With reference to Figure 6, among all graphene samples grown with a laser fluence of $0.9 \, \text{J/cm}^2$ and various cooling rates, the sample that was cooled down at a rate of $10 \, ^\circ\text{C/min}$ shows the highest resistance. This is a surprising result and lacks explanation. A possible explanation could be that $10 \, ^\circ\text{C/min}$ is a critical cooling rate that will affect the crystallization of carbon atoms during the formation of graphene thin films.

  All samples are nanocrystalline multilayer graphene, and their resistance depends on the grain size and affects the FWHM of the G peak of the samples' Raman spectrum [10]. The FWHM of all graphene samples are presented in Figure 7. Comparing the change of FWHM with the cooling rate, laser energy, and the change of resistance, it can be concluded that the resistance and FWHM of the G peak are inversely proportional. This contrasts with the results observed for the graphene monolayer samples, where the broadening of the G peak is associated with the smaller grain size and leads to increased resistance. This unusual behavior can be attributed to the multilayer origin of the graphene samples.

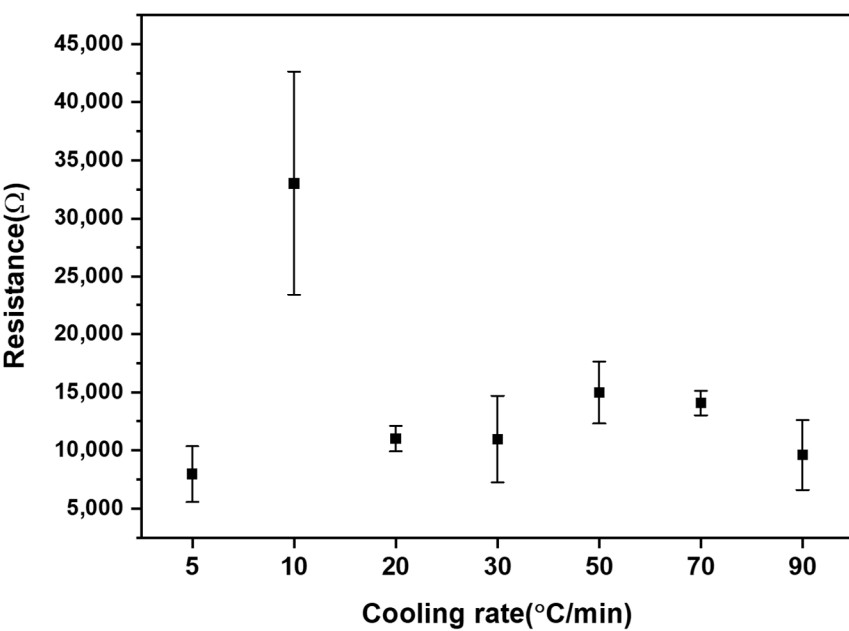

**Figure 6.** The resistance of graphene grown under laser fluence of 0.9 J/cm$^2$ at a different cooling rates. The resistance of graphene grown under different laser energy at the cooling rate of 50 °C/min. The points in the figure are the mean value of the resistance of graphene samples measured while the error bars are the standard deviation of the resistance after multiple measurements on each sample.

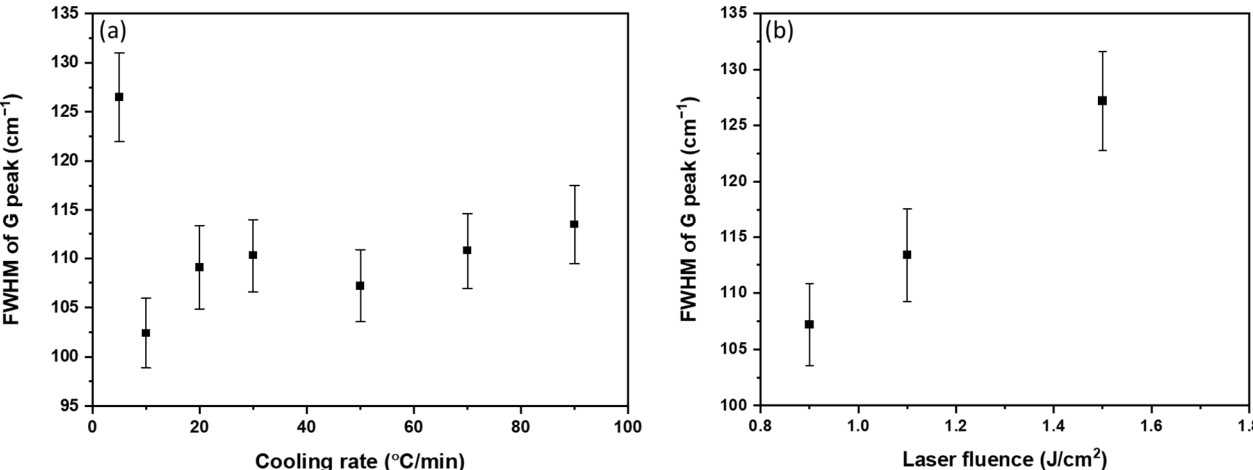

**Figure 7. (a)** The FWHM of graphene samples grown under 0.9 J/cm$^2$ at different cooling rates and **(b)** graphene samples grown at the cooling rate of 50 °C/min under different laser fluence. The trend of the FWHM change of graphene samples is exactly opposite to that of the resistance of the same graphene sample. The points in (a) represent the FWHM of the G peak of Raman spectra of graphene samples grown at different cooling rates and the error bars are the standard deviation due to the FWHM estimation. The points in (b) represent the FWHM of G peak of Raman spectra of graphene samples grown at different laser fluence and the error bars are the standard deviation during the FWHM estimation. To evaluate the relationship between the FWHM of the G peak and the resistance of the multilayer graphene samples, their values were plotted (see Figure 8). The equation of the fitting curve representing the experimental results is shown in Equation (2).

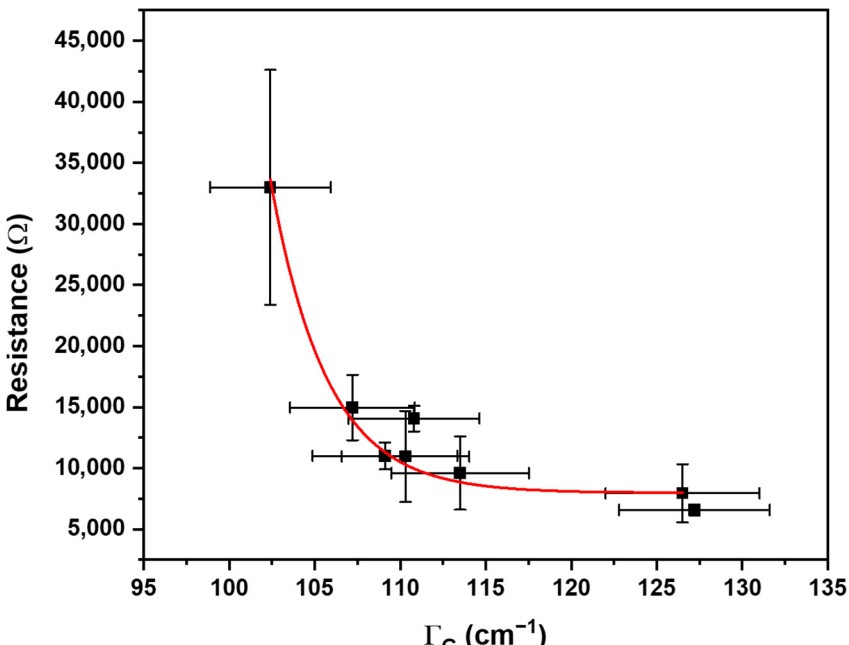

**Figure 8.** The plot between FWHM of G peak and electrical resistance of graphene samples grown under different laser energy and cooling rate. A fitting curve can be plotted to estimate the resistance of the graphene sample with a certain FWHM. The points in the figure are the mean value of resistances of all the graphene samples and the error bars are the standard deviation after multiple measurements of resistance of multiple samples and the estimation of the FWHM of G peak in the Raman spectra of graphene samples.

Based on the data in Figure 7, it can be concluded that the FWHM of the G peak of the graphene samples increases with the increase in the laser fluence. This means that the grain size of the graphene becomes smaller. However, a high laser fluence leads to a low resistance and nanocrystalline structure of the graphene.

$$R(\Omega) = R_0 + A\exp\left(-\left(\Gamma_G\left(\mathbf{cm}^{-1}\right) - \Gamma_{G0}\right)/t\right) \tag{2}$$

where $R_0 = 7.7 \pm 1.1$ k$\Omega$, $\Gamma_{G0} = 102.5$ cm$^{-1}$, A = 25,492.3 $\Omega$, and t = 3.4 $\pm$ 0.6 cm$^{-1}$. All the coefficients are calculated empirically to provide the best fitting. $R_0$ represents the resistance of the graphene of which the FWHM of the Raman G peak is not increased, and t represents the minimal measurable FWHM of the Raman peak. A detailed statistical analysis requires further investigation and more experimental data.

## 4. Conclusions

The present study reveals that the PLD method is suitable to produce nanocrystalline multilayer graphene. It was found that the minimum number of laser pulses that are required to produce fully covered (uninterrupted) samples is 500. This corresponds well with studies reported elsewhere (e.g., S.C Xu et al. [11]). This number of laser pulses resulted in samples that contain ~5 layers of graphene. This result was confirmed by both the Raman and AFM measurements. The number of layers was not affected by the laser fluence and the sample cooling rate after the deposition.

Based on the electrical measurements, it can be concluded that the electrical resistance of the graphene samples is mainly affected by the laser fluence during the depositions and is almost independent of the cooling rate after the deposition. The resistance of the graphene samples decreases from 15 k$\Omega$ to 6.6 k$\Omega$ when the laser fluence increases from 0.9 J/cm$^2$ to 1.5 J/cm$^2$. This suggests that the resistance of the graphene could be further

reduced by increasing the laser fluence; however, Xu et al. report that the quality of the graphene deteriorates when the laser fluence is higher than 6 J/cm$^2$.

Usually, a decrease in the grain size of the graphene is manifested by the increase in the FWHM of the G peak of the Raman spectrum which will lead to an increase in the resistance of grapheneIn our case, this relationship does not hold (see Figure 8). A possible explanation for this could be that the grain size is not the only factor contributing to the resistance of the multilayer graphene grown using the PLD method.

**Author Contributions:** Y.W., P.K.P. and N.A. conceived and designed the research. Y.W., B.Z. and B.R. carried out the experiments. All authors contributed to the paper discussions and manuscript drafting. All authors have read and agreed to the published version of the manuscript.

**Funding:** This work was partly supported by the Henry Royce Institute through the EPSRC grant EP/R00661X/1.

**Data Availability Statement:** The datasets generated during and/or analyzed during the current study are available from the corresponding authors upon reasonable request.

**Conflicts of Interest:** The authors declare no conflict of interest.

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
