# Peer review of "Deposition of Nanocrystalline Multilayer Graphene Using Pulsed Laser Deposition"

_crystals, doi:10.3390/cryst13060881_

Round 1

Reviewer 1 Report

The paper:  “Deposition of Nanocrystalline Multilayer Graphene Using  Pulsed Laser Deposition”, is dealing with an industrial state of the art subject. The authors made a lot of efforts to obtain excellent experimental results. It is a nice success taking into account that PLD is not quite a new physical method.  I consider that the paper could be publishing with a small amendment. The reason is equation (2) which seems not satisfy the unit measure rules.
For example I believe that A should be measure in Ώ and not to be unit less?! In rest the paper is a good one.

Author Response

We thank the reviewer for the useful comments. They help us a lot to improve the manuscript. We have altered its abstract and the introduction part to clarify the outlined issues.

We thank the reviewer for the useful comments. The problem of the unit in equation (2) has been corrected

Reviewer 2 Report

The manuscript reports on the possibility to deposit nanocrystalline multilayer graphene using pulsed laser deposition. The graphene films on Si/SiO2 have been characterized by different analytical techniques and electrical properties have been also evaluated. The paper is surely interesting and the obtained results are valid but an improvement of different sections should be provided. The paper contains useful information in the graphene film field and for these reasons, I consider the paper suitable for publication, but before this, major revisions are needed. In general the paper in the present form is more similar to a technical report than to a scientific paper. I suggest to deepen the interpretation of the obtained results also by comparing the obtained data with those already reported in the pertinent literature. Also the bibliography should be enlarged.

Entire manuscript:

-          The manuscript is nicely written and it is easy to follow, but a linguistic check should be performed (check verbs and plurals).

Introduction:

-          The authors should enlarge the description of the state of the art about graphene film deposition by laser techniques to better highlight the strength of the proposed method. Also the aim of the work should be improved.

Experimental section:

1.      Provide more information about the type of graphite used.

2.      Enlarge the description of the laser deposition procedure reporting all the conditions explored (some details are reported in sections 3 and 4 and not in section 2).

3.      Provide more details about of XPS instrument (brand, model, software, type of data analysis performed)

4.      Provide more details about the AFM measurements (contact or not contact)

5.      Which is the reproducibility of the deposition procedure? Please add some comments on this aspect.

Results and discussion section:

1.      Section 3 should be named as “Results and Discussion” while section 4 should be named as “Conclusions”

2.      Avoid to report comments on the data in the captions.

3.      Describe and comment the different attempts to identify the better conditions to deposit microcrystalline graphene.

4.      Lines 73-75: this information is already present in the experimental section

5.      Report, also as supporting material, the Raman spectra of the samples obtained at different cooling rates (lines 105-109).

6.      I suggest to move the description of XPS data before Raman data to do not separate the description of all the information obtained from Raman spectra.

7.      Figure 6. Remove the panel b. The two numbers are reported in the text and the graphic with only two points is not meaningful.

8.      Add comment and details on the data reported in figure 8.

9.      A comparison with the properties reported in the literature for graphene films obtained by other laser deposition should be reported.

In my opinion, the quality of English language is good. Only minor editing of English language is required

Author Response

We thank the reviewer for the useful comments.

The linguistic of the text has been corrected.

We enlarge the description in the introduction.

The information about graphite target has been added.

The description of laser deposition procedure has been enlarged

The detail of XPS have been added.

The details about AFM measurement has been added

Comment about the reproducibility have been added.

 The title of the corresponding section has been corrected.

The corresponding figure caption has been corrected.

The comments about microcrystalline have been added in the text.

The repeating information has been deleted

The Raman spectra of all the samples are included in both the supporting material and the manuscript

The XPS data has been moved before Raman data.

The panel b in the figure 7 has been removed

Comments about data in Figure 8 has been added.

The comparison between other literature has been added in the introduction.

Round 2

Reviewer 2 Report

The revised version of the manuscript is more complete than the previous one. The Authors improved the manuscript following my comments and advises as well as those of the other Reviewers. The manuscript can be accepted in the present form, anyway I suggest to go through the text checking for language errors and typos still present (check figure numbering and plurals).

 I suggest to go through the text checking for language errors and typos (check figure numbering and plurals).